# Human Papillomaviruses and Epstein–Barr Virus Interactions in Colorectal Cancer: A Brief Review

**DOI:** 10.3390/pathogens9040300

**Published:** 2020-04-20

**Authors:** Queenie Fernandes, Ishita Gupta, Semir Vranic, Ala-Eddin Al Moustafa

**Affiliations:** 1College of Medicine, QU Health, Qatar University, Doha 2713, Qatar; queenie.fernandez@gmail.com (Q.F.); ishita.gupta@qu.edu.qa (I.G.); 2Biomedical Research Centre, Qatar University, Doha 2713, Qatar

**Keywords:** EBV, high-risk HPV, oncovirus cooperation, colorectal cancer

## Abstract

Human papillomaviruses (HPVs) and the Epstein–Barr virus (EBV) are the most common oncoviruses, contributing to approximately 10%–15% of all malignancies. Oncoproteins of high-risk HPVs (E5 and E6/E7), as well as EBV (LMP1, LMP2A and EBNA1), play a principal role in the onset and progression of several human carcinomas, including head and neck, cervical and colorectal. Oncoproteins of high-risk HPVs and EBV can cooperate to initiate and/or enhance epithelial-mesenchymal transition (EMT) events, which represents one of the hallmarks of cancer progression and metastasis. Although the role of these oncoviruses in several cancers is well established, their role in the pathogenesis of colorectal cancer is still nascent. This review presents an overview of the most recent advances related to the presence and role of high-risk HPVs and EBV in colorectal cancer, with an emphasis on their cooperation in colorectal carcinogenesis.

## 1. Introduction

Colorectal cancer is one of the most prevalent types of cancers worldwide [1] that is known to progress gradually over long periods of time [2]. Genetic factors and familial history are considered the principle causative factors for the disease, in addition to environmental factors such as obesity, smoking and consumption of alcohol and red meat [3]. Advanced age is also linked to the onset of colorectal cancer, with incidences increasing sharply between the ages of 30 to 50 years [4]. However, many recent studies have highlighted infectious agents such as oncoviruses as high-risk factors for the disease [5].

As the number of malignancies linked to oncoviruses increases, it is estimated that at least 20% of cancers are attributed to viral infections [6]. To date, a small number of viruses are associated with both solid and non-solid malignancies in humans. The most common oncoviruses include Epstein–Barr virus (EBV), hepatitis viruses B and C (HBV and HCV), human herpes virus 8 (HHV8, also known as Kaposi’s sarcoma-associated herpesvirus) and human papillomaviruses (HPVs) [7,8].

Today, around 150 types of HPVs are recognized and identified as either high- or low risk viruses. There are at least 17 high-risk types (16, 18, 31, 33, 35, 39, 45, 51, 52, 55, 56, 58, 59, 68, 73, 82 and 83); it is believed that persistent infection with high-risk HPVs may lead to malignancies in cooperation with other oncogenes [9,10]. On the other hand, low-risk HPVs (6 and 11) are not linked to cancer, and their infections result in the development of benign gynecological papillomas and skin warts [11,12]. HPVs-16 and 18 account for around 70% of cervical cancer cases, as well as other cancers of the anogenital tract, e.g., the vulva, vagina, penis and anus [13]. However, recently HPVs have been found to be associated with cancers of non-sexual regions like colon, rectum, esophagus, breast, skin, bladder and head and neck cancers [9,14,15,16,17]. HPV as an infectious agent can infect epithelial cells (like those of the colon), subsequently triggering carcinogenesis [18]. However, according to a few studies, the link between HPV and colorectal cancer is still controversial [19,20,21,22]; other studies have shown almost 40%–80% of colorectal and 30% of head and neck cancer cases are positive for high-risk HPVs [21,22,23,24,25].

On the other hand, EBV—a member of gamma-herpes viruses—is one of the first oncoviruses to be studied in human carcinogenesis [26]. The primary mode of transmission of the virus is through the saliva [27]. EBV is ubiquitously present among human adult populations, with an estimated prevalence of over 90% in individuals by the age of 35 years. EBV is known to cause infectious mononucleosis, as well as cancers such as Hodgkin’s, several subtypes of non-Hodgkin lymphomas [(B and T-cell, Natural Killer (NK)], as well as gastric and nasopharyngeal carcinomas [28,29,30]. Moreover, EBV has also commonly been linked to several other cancers such as breast, cervix, prostrate, oral cavity and salivary glands [10,31,32,33,34].

Currently, the role of HPVs and EBV in the pathogenesis of colorectal cancer is still ambiguous. In addition, reports detailing their oncogenic activity in colorectal cancer are sparse. Here we attempt to review and summarize the presence/co-presence and role of high-risk HPVs and EBV in the pathogenesis of colorectal cancer.

### 1.1. Human Papillomaviruses (HPVs) and Their Role in Colorectal Cancer

HPVs are non-enveloped double-stranded DNA viruses that are capable of infecting the epithelial cells of the skin and mucosa. The genome of HPVs consists of an 8-kb circular DNA encased in a capsid shell that is composed of a major (L1) and minor (L2) capsid proteins. The HPV viral genome codes for both early (E1, E2, E4, E5, E6 and E7) and late (L1 and L2) proteins that are critical for host infection [35,36].

HPV-inflicted carcinogenesis is a multistep process. It often begins through the primary infection of the proliferating epithelium. The virus maintains its DNA at low copy numbers in the infected basal cells of the host. However, during differentiation of epithelial cells, HPV displays its virulence by replicating to a high copy number, thus expressing the capsid envelope genes (L1 and L2) resulting in virion production that are subsequently released from the epithelial surface [37] spreading infection and leading to a hyperproliferative state [38]. Almost all high-risk HPV types prevent host immune recognition and promote persistent infection leading to neoplastic transformation. Moreover, the HPV cycle is entirely intraepithelial, non-lytic and inhibits activation of pro-inflammatory signals, leading to the recruitment of antigen presenting cells, followed by the subsequent release of cytokines triggering growth and proliferation [37].

In addition to binding and inhibiting the activity of tumor suppressor molecules like p53 and pRB [39,40], the E6/E7 oncoproteins display alternate mechanisms of inflicting oncogenesis. E6 functions to cause telomerase activation, causing deregulation of pathways involved in cellular proliferation, differentiation, immune recognition and survival signaling [37]. In contrast, E7 enhances genomic instability, thereby resulting in the accumulation of chromosomal abnormalities [41]. This cell-cycle deregulation, telomerase activation and induced genomic instability creates a favorable environment for neoplastic transformation of cells. Furthermore, HPVs can also trigger oncogenesis through the inactivation of the E2 gene, the main inhibitor of E6/E7 oncoproteins [42].

In recent years, a number of studies have attempted to examine the relationship between HPVs and colorectal cancer. Most of these studies are based on utilizing polymerase chain reaction (PCR) analyses to determine the presence of high-risk viral DNA in fresh and formalin fixed paraffin embedded tumor tissue. As mentioned above, high-risk HPVs have carcinogenic effects in around 40%–80% and 30% of colorectal as well as head and neck, respectively [21,22,23,24,25], especially in their invasive forms [43]. Several studies have highlighted the presence of high-risk HPVs (HPVs-16, 18, 31, 33 and 35) in human colorectal cancers [44,45,46,47]. A meta-analysis has also confirmed the presence of high-risk HPVs in colorectal cancers [48]; however, the prevalence of high-risk HPVs differs from one geographic location to another [49,50].

While studies carried out from the Middle East region, such as Iran [51,52,53,54] report low-to-medium frequencies of high-risk HPV types, in contrast, studies carried out in other parts of the world [14,24,55,56,57,58], report medium-to-high prevalence of high-risk HPV infections in adenocarcinoma colorectal tissue, in comparison to normal tissue. The most frequently expressed HPV types found in colorectal cancer include HPV 16, 18 and 33 [59]. Our group reported a frequency of around 54% of high-risk HPVs in colorectal samples in the Syrian population [44], and established an association between high-risk HPVs and invasive cancer phenotype. We also demonstrated that high-risk HPV type 16 oncoprotein converts non-invasive and non-metastatic human cancer cells into invasive and metastatic forms [9].

The underlying mechanism of HPV in cancer pathogenesis involves integration of its pro-viral DNA into host DNA resulting in direct inactivation of the E2 gene [23]. The E2 gene is a negative regulator of the E6/E7 oncoproteins that bind to tumor suppressor molecules like p53 and pRB respectively, thus disrupting their tumor suppressing activity [60]. This phenomenon is known to mark oncogenesis that further leads to genomic instability and consequently cellular transformation [61,62]; indicating mechanism of HPV-induced colorectal carcinogenesis.

In another study, we reported that E6/E7 oncoproteins of high-risk HPV type 16 cooperate with the ErbB-2 receptor to provoke cellular transformation of normal epithelial cells [63]. We also found that D-type cyclins (D1, D2 and D3) are elemental for cellular transformation induced by E6/E7/ErbB-2 cooperation [63,64,65] via β-catenin tyrosine phosphorylation through pp60 (c-Src) kinase activation [66,67,68]. Additionally, we found that E6/E7 of HPV type 16 provoke cell-invasive and metastatic abilities in vitro and in vivo, accompanied by Id-1 overexpression, which regulates cell invasion and metastasis of cancer cells [63]. To determine the role of high-risk HPVs in colorectal cancer, we assessed the effect of E6/E7 of HPV type 16 in two human primary normal colorectal “mesenchymal” cell lines (NCM1 and NCM5), established in our laboratory [69]. E6/E7 oncoproteins stimulated the upregulation of D-type cyclins and cyclin E as well as Id-1 to enhance cell proliferation, transformation and migration [65]. While it is important to state that the role of E5 oncoprotein in these malignancies still lies nascent, it has been reported that E5 of high-risk HPVs can affect cell alteration and consequently lead to oncogenesis via its interaction with EGF-R1 pathways, MAP kinase and PI3K-Akt, as well as pro-apoptotic proteins [70,71,72].

In general, the expression of E6 in cancer cells is associated with overexpression of Fascin, Id-1 and P-cadherin— all of which are actively involved in cell invasion and metastasis [44,73,74,75]. Also, it has been pointed out that E5 and E6/E7 oncoproteins can cooperate in cancer progression via the EMT event [76]; indicating the potential cooperation of HPVs oncoproteins in the progression of human cancers including colorectal.

Several other studies reveal that high-risk HPVs may also lead to the activation of oncogenes like KRAS and C-MYC. HPV DNA integration is often found in fragile sites around regions of the C-MYC locus [60,61,62,77,78]. C-MYC belongs to the multi-genic family of Myc, and is located on chromosome 8q21. It encodes for a transcription factor that is largely involved in differentiation, apoptosis and regulation of the cell cycle [79]. C-MYC is activated through the deregulation of DNA expression, followed by gene amplification that increases the production of excess gene copies, which is a hallmark of most malignancies, often known to increase with tumor grade [80,81,82]. Similarly, the proto-oncogene, KRAS—located on chromosome 12—encodes for a protein involved in proliferation, differentiation as well as signal transduction. Its pathogenesis in colorectal cancer is linked to several mutations at codons 12, 13 and 61 that lead to uncontrolled proliferation [83]. KRAS mutations are responsible for the transition of adenomas to carcinomas [84,85]; and are present in 56% of HPV positive tumors [46]. However, an earlier study showed no association between KRAS and C-MYC activation and infection through HPVs [46]. Therefore, roles of oncogenes like C-MYC and KRAS need to be further analyzed in order to understand the underlying mechanism of HPV-associated colorectal cancers.

Moreover, it has been revealed that tumors escape recognition by the immune system through the acquisition of apoptosis resistance that may occur due to the downregulation of pro-apoptotic molecules [86]. E6/E7 viral oncoproteins inhibit host-triggered apoptosis either through the inactivation of p53 or via the downregulation of TNF-R1 that leads to the disruption of apoptosis [87]. Studies have confirmed downregulation of two pro-apoptotic genes in particular, TNFRSF6 and DR5 in HPV-associated colorectal cancers [88,89].

Earlier studies have shown that HPV oncoproteins may play a co-stimulatory role with commonly known colorectal cancer mutations. However, one study shows a lack of association between HPV status and microsatellite instability that is a common biomarker of colorectal cancer [90]. Nevertheless, the role of HPV oncoproteins in cancer, particularly in colorectal cancer is not fully elucidated.

### 1.2. Epstein Barr Virus and its Role in Human Colorectal Cancer

The Epstein–Barr virus is a DNA virus that belongs to the family of the herpes virus and was first identified in a Burkitt lymphoma cell line. The virus was found to be distinct from other viruses including herpes simplex, herpes zoster or cytomegalovirus [91,92]. Its genome is represented by a double-helix DNA of approximately 1.1 × 10^8^ Daltons in size [93] that mainly infects the epithelial as well as B-cells, often leading to malignancies. Infection is brought about through the fusion of the viral and cellular lipid bilayer membranes, involving complex mechanisms of multiple viral factors and host receptors.

The presence of EBV is found to be ubiquitous. Nearly 90% of the human adult population is infected by the virus [94,95]. Moreover, lifelong persistence is often the hallmark of most herpesvirus infections. EBV infections occur typically during early childhood. However, most of them are termed mild infections. While infections that occur during early adulthood are linked to infectious mononucleosis, this disease is defined by a triad of symptoms including fever, lymphadenopathy and pharyngitis [95].

The virus exhibits dual tropism; it is capable of infecting both B and epithelial cells [96] by alternating its envelop proteins [97]. EBV infection is commonly associated with B-cell lymphomas (Burkitt and Hodgkin lymphoma) as well as epithelial malignancies (nasopharyngeal [98], gastric [99] and probably rectal carcinomas [26]). Additionally, it is also associated with T-cell and/or Natural Killer (NK) cell lymphoproliferative disease as well as those found in immunosuppressed individuals (HIV infected or patients who have undergone transplantation surgeries) [100].

During lytic infection, genes encoded by EBV selectively replicate virion components (viral DNA genomes and proteins). However, during latent infection cycles, EBV encodes viral genes including the six nuclear antigens (EBNA-1, -2, -3A, -3B, -3C and LP), three Latent Membrane Proteins (LMP-1, -2A and -2B), two small non-coding Ribonucleic acids (EBER-1 and -2) as well as the BamHI-A rightward transcripts [101]. The main function of these EBV proteins is to help maintain viral existence by evading the natural mechanisms of immune surveillance [102]. In addition, it is established today that these genes can play an important role as “oncogenes” in infected cells.

EBV exerts a number of immune-suppressive effects that aid in carcinogenic transformation such as silencing the anti-EBV effect of interferon-gamma (INF-γ) in B cells and modulating the production of anti-viral cytokines like TNF-α, IL-1β and IL-6 [103]. Furthermore, EBV is capable of mimicking the characteristics of IL-10 thereby permitting its escape from the host’s anti-viral response [103,104]. On the whole, a compromised immune system and a chronic inflammatory microenvironment enhances the oncogenic properties of EBV [105].

When it comes to colorectal cancer, several studies have established a causative link between EBV and colorectal carcinogenesis [106,107,108]. The presence of viral EBV DNA was detected in tumor samples through the utilization of techniques like in situ hybridization (ISH), Immunohistochemistry (IHC) and Polymerase Chain Reaction (PCR)-based methods [26]. These techniques may have different sensitivities for detecting EBV targets (genes or proteins) which may partially explain discrepancies in the percentage of EBV positivity that has been reported in current literature (0%–46%) (Table 1). IHC and ISH assays typically detect EBNA or LMP family of proteins all of which can be readily visualized by a microscope (LMP proteins are cytoplasmic/membranous while EBNA proteins are nuclear products). In case of PCR-based assays, the detection of viral load by specific primers may be substantially affected (contaminated) by the EBV load within inflammatory cells (lymphocytes) that frequently accompany colorectal cancer samples [26].

EBV has also been associated with nasopharyngeal carcinomas, a distinct subtype of head and neck carcinoma with a prominent lymphoid component. Also, a particular type of primary colorectal cancer [119], lymphoepithelioma-like carcinoma (LEC) is morphologically similar to nasopharyngeal carcinoma [120] and is often associated with EBV infection [121,122,123]. Moreover, although rare, non-epithelial colorectal malignancies are also associated with EBV infection. Most of these are often hematological malignancies like Hodgkin’s or non-Hodgkin’s lymphomas originating from B-cells, T-cells or NK-cells. In addition, Burkitt lymphoma has also demonstrated the presence of EBV. Interestingly, ISH and IHC assays revealed that EBV may be linked with the onset of colorectal cancer in post-transplant patients [26,124,125,126,127]. Notably, studies conducted by three different groups [124,126,128] also analyzed EBV status in smooth muscle tumors of the colon using ISH, IHC and PCR. Although, the number of cases was small in these investigations, all three of them reported the presence of EBV in 100% of their population [124,126,128]. Worldwide, studies using RT-PCR and IHC assays have reported a prevalence of 20%–50% for EBV in colorectal cancer [116,129,130,131,132]. Although, reports pertaining to the presence of EBV in the Middle East are sparse; nevertheless, studies from Syria and Iran used PCR and revealed the presence of EBV in 36% [133] and 38% [116] of the cases, respectively. A particular investigation using methylation-specific PCR reported that although EBV infection was found in 19% of colorectal cancer cases, viral DNA showed no association with the methylation of thirteen cancer-related CpG islands that were addressed [134].

Table 2 briefly summarizes worldwide EBV prevalence in human colorectal cancers in different populations.

On the other hand, recent studies on colorectal and related gastric cancers have focused on a new tumor suppressor gene *ARID1A* (*AT-rich Interactive Domain-containing 1A protein*) [146,147,148,149,150,151,152,153,154,155,156]. The *ARID1A* gene encodes a large nuclear protein involved in the regulation of several cellular processes including cell differentiation and DNA repair [157]. *ARID1A* is mutated in colorectal and gastric cancers [146,147,148,149,150,151,152,153,154,157]. Interestingly, data pooled by a meta-analyses on colorectal and gastric cancer confirmed the presence of EBV infection [158] as well as reported loss of ARID1A protein expression associated with advanced grade and tumor differentiation [158]. However, further investigations are required to address the mechanism of interaction between EBV and mutation in *ARID1A* to gain a better understanding of their combined effect in initiating/mediating tumor progression.

In addition, a number of studies have reported the role of *Fascin* gene in the progression of several cancers including colorectal [159,160,161] as *Fascin* is often over-expressed in several types of invasive cancers [73,162,163,164]. Presence of EBV in colorectal cancer is frequently accompanied with an over-expression of *Fascin* [133]. Our previous study showed that expression of EBV oncoprotein (LMP1) correlates with *Fascin* overexpression and an invasive form of colorectal cancer (moderately to poorly differentiated adenocarcinomas) [133]. Earlier investigations reported that LMP1 and EBNA1 oncoproteins of EBV enhance cancer progression and metastasis of nasopharyngeal carcinoma through the initiation of the epithelial-mesenchymal transition (EMT) event [165,166,167]. Initial studies showed LMP1-mediated over-expression of *Fascin* is dependent on *NF-κB* as both, *Fascin* and *NF-κB,* aid in invasion and migration of LMP-1 expressing lymphocytes [168]. Similar to cancer progression in nasopharyngeal carcinoma, we postulated that the presence of EBV in colorectal cancer can promote cancer progression through the initiation of EMT event via EGF-receptor and/or Akt-signaling pathways as well as Wnt/β-catenin-signaling [166,169,170]. Thus, it is clear that EBV may play an important role in colorectal cancer progression; however, more comprehensive studies are necessary to test these hypotheses and determine the role of EBV in colorectal cancer.

### 1.3. The Interaction of High-Risk HPVs and EBV in Human Cancers

Epithelial cells can be co-infected with more than one viral species including HPVs and EBV [171,172,173]. Various studies showed co-infection of HPVs and EBV in different cancers including cervical, oral as well as breast [28,34,174,175,176,177,178]. HPVs and EBV co-infections may have a major role in the initiation and/or progression of cancer [175]; co-infection with more than one type of oncoviruses is necessary to attain complete transformation [10,64,65,171]. As stated above, E6/E7 onco-proteins of HPV 16 cooperate with HER-2 to provoke cellular transformation and initiate EMT of human normal oral epithelial cells [65,171]. Furthermore, HPVs and EBV can enhance the onset and spread of human carcinomas via the crosstalk of oncoproteins and signaling pathways (β-catenin, JAK/STAT/SRC, PI3k/Akt/mTOR and/or RAS/MEK/ERK pathways), as illustrated in Figure 1 [10,179].

As one of the most potent oncoviruses, HPVs and EBV infect epithelial tissues in a similar pattern that enables them to transform normal cells into malignant ones in cooperation with another oncogene. Both viruses can infect and replicate in upper aerodigestive epithelial cells as well as in the epithelium of the colon and rectum stimulating the productive and lytic phases of the HPV and EBV life cycle, respectively. A study by Makielski et al. revealed that high-risk HPV stabilizes the EBV genome and promotes EBV lytic reactivation in differentiated epithelial cells suggesting that EBV and HPV co-infection elevates EBV-mediated pathogenesis of cancer [173]. Although E6/E7 oncoproteins of HPV are essential to provoke EBV lytic reactivation in the suprabasal layer of oral epithelia, the underlying mechanisms still lies nascent.

Moreover, studies also state that EBV may play an indirect role in promoting the oncogenic pathogenesis of HPV by inhibiting natural immune responses directed towards HPV-transformed cells. This can take place through the production of the viral BCRF1 gene product, which is an interleukin-10 homolog [180,181]. If EBV gene products are secreted in the exosome, cells infected with EBV may affect tumor-tumor microenvironment, thus leading to the suppression of immune responses towards HPV [182,183]. Studies have also shown that the presence of EBV may also enhance the genomic instability of HPV-infected epithelial cells thereby further promoting the progression of cancer [184,185].

Interestingly, other than lytic replication, oncogenic viruses can enhance cancer by promoting immunosuppression in human virus-associated cancers and triggering DNA damage response [186,187]. Lately, the apo lipoprotein B mRNA editing enzyme catalytic polypeptide 3 (APOBEC3) was found to stimulate intrinsic immunity from several pathogens, including viral infection [188,189]. APOBEC3 can excise both HPV and EBV by stimulating cytidine-to-uracil mutations in viral DNA [190]. While in breast cancer, HPV-18 stimulates APOBEC3B activity—leading to genome instability [191]—in cervical cancer, beta interferon prompt E2 hypermutation in HPV-16 by APOBEC3 [192]; thus, suggesting viral infection results in APOBEC3 induced integration of viral DNA in the host genome via genome instability (Figure 1).

HPVs and EBV co-infections promote invasion ability of human oral cancer [193] as well as breast cancer [172,176,194,195]. While HPVs and EBV were detected in 15%–20% of oral SCCs [178,196], HPVs and EBV co-prevalence in tonsillar and tongue SCCs was 25% and 70%, respectively [197]. In our previous study using Syrian breast cancer samples, we found 32% of the cases were positive for both high-risk HPVs and EBV; co-infection was associated with high-grade invasive ductal carcinomas and lymph node involvement [10]. Furthermore, in cervical cancers, HPVs and EBV were co-present in approximately 29% of the cases and were correlated with an invasive cancer phenotype [28,198]. On the other hand, in prostate cancer co-presence of HPV and EBV was significantly higher (55%) than normal and benign tissues [32]. Additionally, experimental evidence show that HPV and EBV work together to promote the proliferation of cultured cervical cells [199], suggesting the same may be true for prostate epithelial cells. Although, studies examining co-presence of HPVs and EBV in colorectal cancers were performed [130,134], their co-presence was rarely detected [131]. A study performed by our group analyzed one-hundred and eight rectal cancers for the co-presence of EBV and HPV. Based on our findings, the co-incidence of these oncoviruses was detected in 11% of the samples [91]. We also recently reported that the co-presence of high-risk HPVs and EBV in colorectal cancer correlates with tumor aggressiveness in Syrian colorectal cancer patients [92].

Various plausible mechanisms have been proposed to further elucidate the mechanism of co-infection with HPVs and EBV in the pathogenesis of cancers originating from the epithelia. Increase in HPV results in loss of microRNA, miR-145 [200] which down-regulates KLF4 expression. Thus, HPV has shown to elevate the expression of KLF4, which further results in EBV lytic reactivation [201]. A study by Makielski et al. showed that EBV alone stimulates E2F-responsive protein expression; thus, indicating that EBV reprograms lethally differentiating cells to support cell cycle progression by HPV oncogenes [173]. Since HPV increases the capacity of epithelial cells to support the EBV life cycle during the lytic phase, EBV accumulation in epithelial cells may increase malignancy [173]. While stimulation of the lytic cycle in cells can also elevate the expression of different viral and cellular cytokines resulting in increased cellular differentiation by activating signaling pathways including the protein kinase R, mitogen-activated protein kinase (MAPK) pathways and NF-κB [202,203,204]. In addition, other studies state that infection through EBV can boost the invasive properties displayed by epithelial cells expressing E6 and E7 oncoproteins of HPV [179]. This further confirms that EBV may be responsible for the rapid progression of EBV/HPV–related cancers. Therefore, based on the various roles of HPVs and EBV in cancer pathogenesis, we postulate that oncoproteins of HPVs can interact with those of EBV (LMP1 and/or EBNA1) and result in progression and metastasis by enhancing the EMT event of different types of cancers including colorectal (Figure 1) [179].

## 2. Conclusions

Oncovirus-associated cancers are now becoming a worldwide concern. It has become evident that HPVs and EBV can be co-present in several types of human carcinomas including colorectal. Thus, the role of HPVs and EBV co-infection in cancer development should be further addressed including the epigenetic role of this cooperation, which can help to understand the underlying mechanism of this co-infection in the onset and development of malignant tumors. Additionally, chromatin control of viral co-infection also exemplifies a new field with candidate targets for the development of novel antiviral therapies.

The current research indicates a potentially plausible relationship between the co-presence of HPV and EBV in the pathogenesis of colorectal cancer. The functional roles of both viruses span over the beginning of oncogenesis or neoplastic transformation to tumor progression, and finally, the attainment of metastatic properties. The intricate mechanisms through which the viruses escape immune recognition are complex multi-stage processes that still need to be studied in considerable depth. However, further studies involving both translational and clinical aspects in a larger cohort are required to elucidate the oncogenic importance of their co-presence and its clinical impact.

Large-scale functional genomic analyses have previously identified viral lytic genes co-expressed with cellular cancer-associated pathways, indicating that the lytic cycle plays a role in virus-mediated oncogenesis [205,206]. Further research using genome-wide approaches can pave the way for the development of inclusive models of persistent HPV and EBV interactions and its underlying roles in infected cells.

Finally, understanding the mechanisms through which the viruses sustain and promote shared virulence is a major step towards developing therapeutic strategies in oncoviruses-related cancers. Meanwhile, EBV and HPVs vaccines, upcoming and available, respectively, can be used as a preventive strategy against infections with these oncoviruses and their associated cancers.

## Figures and Tables

**Figure 1 pathogens-09-00300-f001:**
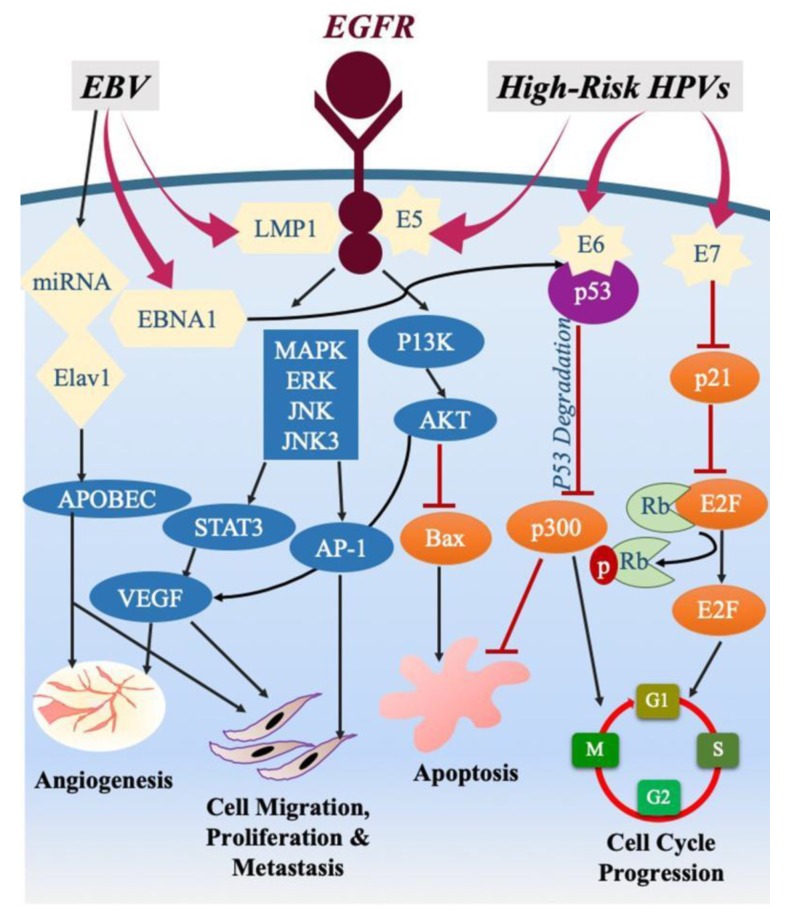
Schematic outline showing plausible crosstalk between the oncoproteins of high-risk human papillomaviruses (HPVs) and Epstein–Barr virus (EBV) in the induction of angiogenesis, cancer progression, apoptosis as well as cell cycle progression. We note that high-risk HPVs and EBV oncoproteins share various downstream-signaling pathways including MAPK/ERK/JNK/JNK3, PI3k/Akt and p53/p21 as well APOBEC; HPVs/EBV oncoproteins can closely cooperate in cancer development and/or enhance cancer progression.

**Table 1 pathogens-09-00300-t001:** Human papillomaviruses (HPV) prevalence in human colorectal cancers in different populations around the world.

Population (Year)	Number of Samples	HPV Status (%)	Assay (Detection Method)	References
Bosnian (2020)	106	Positive (50%)	PCR and IHC	[109]
Syrian (2020)	102	Positive (37%)	PCR and IHC	[110]
Portuguese (2020)	144	Negative	RT-PCR	[111]
Polish (2017)	50	Positive (20%)	PCR	[112]
Puerto Rican (2016)	45	Positive (42%)	PCR	[23]
Brazilian (2016)	1,549	Positive (52%)	Meta-analysis	[113]
Syrian (2012)	78	Positive (54%)	PCR and IHC	[44]
Turkish (2011)	106	Negative	PCR	[47]
Argentinian (2010)	75	Positive (44%)	PCR	[114]
Israeli (2010)	106	Negative	RLB and LiPA	[19]
USA (2010)	73	Negative	RLB and LiPA	[19]
Spain (2010)	100	Negative	RLB and LiPA	[19]
Turkish (2009)	56	Positive (82%)	PCR and southern blot hybridization	[45]
Italian (2008)	66	Positive (33%)	PCR	[115]
Brazilian (2007)	72	Positive (83%)	PCR	[21]
Turkish (2006)	53	Positive (81%)	PCR	[46]
USA (2005)	55	Positive (51%)	PCR	[116]
Argentinian (2005)	27	Positive (74%)	PCR	[117]
USA (1992)	50	Negative	PCR	[118]

IHC: immunohistochemistry; ISH: in situ hybridization; PCR: polymerase chain reaction: RLB: Reverse line blot; LiPA: Line Probe Assay.

**Table 2 pathogens-09-00300-t002:** Epstein–Barr virus (EBV) incidence in colorectal cancer samples in different populations.

Population (Year)	Number of Samples	EBV Status (%)	Assay (Detection Method)	References
Bosnian (2019)	108	Positive (25%)	PCR and IHC	[92]
Iranian (2018)	210	Positive (1.4%)	PCR	[118]
Syrian (2017)	102	Positive (36%)	PCR and IHC	[133]
Iranian (2016)	35	Negative	PCR	[135]
Iranian (2015)	50	Positive (38%)	PCR	[116]
Chile (2015)	37	Positive (46%)	PCR	[136]
Italian (2014)	44	Negative	RT-PCR and IHC	[130]
North America (2013)	117	Positive (21%)	PCR	[131]
Polish (2011)	186	Positive (19%)	PCR	[134]
South Korean (2010)	72	Positive (30.6%)	IHC and ISH	[127]
Japanese (2010)	1	Negative	IHC	[137]
Italian (2009)	100	Positive (2.8–39%)	RT-PCR and sequencing	[138]
Chinese (2006)	90	Positive (30%)	IHC and ISH	[129]
Chinese (2003)	130	Positive (5–8%)	IHC, ISH and PCR	[117]
Scotland (2003)	26	Negative	ISH	[139]
Argentina (2002)	19	Positive (5%)	ISH	[140]
Japanese (2001)	102	Negative	ISH	[141]
South Korean (2001)	274	Negative	ISH	[142]
Chinese (1994)	36	Negative	ISH	[143]
Czechoslovakia (1988)	13	Negative	PCR	[144]
New Guinean (2004)	46	Positive (46%)	ISH	[145]

IHC: immunohistochemistry; ISH: in situ hybridization; PCR: polymerase chain reaction.

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
