# Peer review of "Human Papillomaviruses and Epstein–Barr Virus Interactions in Colorectal Cancer: A Brief Review"

_pathogens, 2020, doi:10.3390/pathogens9040300_

Round 1

Reviewer 1 Report

“Human papillomaviruses and Epstein–Barr virus interactions in colorectal cancer: A concise review”

Summary:

This article reviews the detection of two viruses, Human Papillomaviruses and Epstein-Barr virus (EBV), in colorectal cancer. Both viruses are known cancer-causing and cancer-associated viruses in a number of cell/tissue types. This review focuses on the presence of the viruses in colorectal tumors, a field that is less established than other tumor types. First, the role of human papillomaviruses in colorectal cancer is discussed, including the viral proteins that may be involved and the cellular pathways that may be affected. Then, the role of Epstein-Barr virus in colorectal cancer is described along with similarities to other more well-known EBV-associated tumors. Finally, the possibility of co-infection by both viruses is discussed. The cellular pathways targeted by both viruses, and possible crosstalk is described. Overall, the case is made that increased screening of colorectal tumors is needed to determine the prevalence and role of these oncogenic viruses in colorectal cancer.

Broad comments

This is a review article summarizes the studies completed thusfar on the detection of HPV and EBV in colorectal cancer. The review is thoroughly referenced. The overall organization of discussing each virus alone and then the possible outcomes of co-infection is logical and easy to follow.

  1. The summary table of previous studies of EBV in colorectal cancer is convenient. Could a similar table be generated for HPV?
  2. Is would help to expand the discussion of the assayed used to detect the viruses. (around Line 205)
    What viral genes/proteins are the targets of the assays? Is there any correlation between the method of detection and the data?

Specific comments

  1. Line 170 Burkitt lymphoma

Reviewer 2 Report

This manuscript reviews the roles of HPV and EBV, separately and together, in carcinogenesis, with some emphasis on colorectal cancer.  The possible effects of these two very common viruses on colorectal cancer is an appropriate topic, as there is significant evidence for a role for their interaction in other tissues.  Table 1, a compilation of studies assessing the presence of the EBV colorectal cancers, is a valuable resource for the field. 

Much space is devoted to the discussion of the viruses' roles in non-colorectal tissues.  The manuscript would benefit from clearer organization, including significant editing to clarify the distinctions between what is known from that work on other tissues and what is known for colorectal cancer. 

Specific comments:

Generally:  "On the other hand" is used excessively and confusingly.  Its most common meaning is "by contrast," but it is used here almost exclusively to mean "also."

Title:  All good scientific writing is concise.  "a brief review" or "an overview" would be more appropriate.

Line 22:  While studies suggest that HPV and EBV play a role in some breast cancers, the data do not support the conclusion that these viruses "play a principal role."  Please omit "and breast."

Line 23:  The beginning of this sentence is unnecessary, suggest starting with "Oncoproteins."

Line 27:  Please replace "the concise" with "an" or "a brief"

Line 43:  Please provide the more common name of HHV8, for example "(HHV8, also known as KSHV)"

Lines 48-49:  The statement regarding the inability of HPV infection alone to induce neoplastic transformation is too categorical and is not sufficiently supported by the reference provided (#11).  Please rewrite or omit.

Line 63:   Mononucleosis is not a cancer; please omit "other" from "other cancers"

Line 84:  "promotes" should be singular, "promote"

Lines 80-91:  Reference 38 does not support these statements.

Lines 85-87:   This sentence needs punctuation.

Lines 115-119:  The thesis statement of this paragraph declares that inactivation of the E2 gene is critical to HPV-induced colorectal carcinogenesis.  However, the supporting statements and references are based on work in other tissues.  Please rewrite to clarify that you are speculating that the role of HPV in  colorectal cancer will be found to be analogous to its role in, for example, cervical cancer.

Lines 116-117:  This sentence has grammatical issues.  "The E2 gene, a negative regulator of the E6/E7 oncoproteins bind" should be replaced with "The E2 gene is a negative regulator of the E6/E7 oncoproteins that bind"

Lines 121-123:  The statement that HPV cannot transform on its own is unproven.  HPV's ability to interact with a cellular factor to promote transformation supports, rather than disproves, HPV's ability to transform in the absence of genetic changes.

Line 157:  Please replace "acquirement" with "acquisition"

Line 180:  Those who develop mononucleosis are typically between 15 and 24 years old; recommend describing this as "early adulthood"

Line 183:  "its envelope"

Line 185:  Please provide a reference here, e.g. Bedri et al 2019, to support your speculation that EBV is "probably" causative in rectal cancer.

Line 210:  The colorectal cancer's resemblance to nasopharyngeal carcinoma may have nothing to do with EBV.  Please omit "hence" and provide a reference if LEC's are frequently found to be infected with EBV.

Lines 251-253:  This speculation should be condensed, for example, to "More comprehensive studies are necessary to test these hypotheses and determine the role of EBV in colorectal cancer."

Line 255:  "It is evident that" is unnecessary and should be removed.  Also, please provide a reference supporting the statement that individual cells are infected by both viruses.

Lines 257-259:  This sentence is both poorly written and includes the unproven statement that these viruses cannot induce carcinogenesis individually.  Please omit or rewrite.

Figure 1:  The effect of HPV on APOBEC should be shown in the figure and/or mentioned in the legend as a common target of EBV and HPV.

Line 295:  Do the authors mean "stimulate" rather than "stipulate"?

Line 324:  Please clarify "Since HPV alleviates EBV in epithelial cells,".  How does HPV "alleviate" EBV?

Line 343:  Please omit "Evidently." 

Line 353:  Please clarify "viral persistence and its underlying role in HPV and EBV interaction in the infected cells."  One would think that interactions between HPV and EBV would have an underlying role in viral persistence, rather than the other way around.

Round 2

Reviewer 2 Report

The authors' changes are helpful, and the addition of the compilation of studies assessing the presence of HPV in colorectal cancers (now Table 1) is  valuable. Upon re-reading the manuscript, however, it's clear that the language needs significant improvement.  Many sentences have significant grammatical problems. 

The following should be corrected.  They are related cases where the references do not support the corresponding statement:

listing Lines 54-56:  Please provide a reference supporting the statement that up to 80% of colorectal cancers are infected with HPV.  The references given involve head and neck and anal cancers, not colorectal cancers.

Lines 96-97:  Please provide a reference supporting the statement that up to 70% of colorectal cancers are infected with HPV.  The references given involve head and neck and anal cancers, not colorectal cancers. Please also address the fact that the claim is 70% here, and 80% in lines 54-56.

Author Response

Please see attached, as requested. 
